# Deciphering the Roles & Regulation of Estradiol Signaling during Female Mini-Puberty: Insights from Mouse Models

**DOI:** 10.3390/ijms232213695

**Published:** 2022-11-08

**Authors:** Marie M. Devillers, Sakina Mhaouty-Kodja, Céline J. Guigon

**Affiliations:** 1Sorbonne Paris Cité, Université de Paris Cité, CNRS, Inserm, Biologie Fonctionnelle et Adaptative UMR 8251, Physiologie de l’Axe Gonadotrope U1133, CEDEX 13, 75205 Paris, France; 2Neuroscience Paris Seine—Institut de Biologie Paris Seine, Sorbonne Université, CNRS UMR 8246, INSERM U1130, 75005 Paris, France

**Keywords:** ovary, estradiol, estradiol receptor, hypothalamus, mini-puberty, GnRH

## Abstract

Mini-puberty of infancy is a short developmental phase occurring in humans and other mammals after birth. In females, it corresponds to transient and robust activation of the hypothalamo-pituitary-ovarian (HPO) axis revealed by high levels of gonadotropin hormones, follicular growth, and increased estradiol production by the ovary. The roles of estradiol signaling during this intriguing developmental phase are not yet well known, but accumulating data support the idea that it aids in the implementation of reproductive function. This review aims to provide in-depth information on HPO activity during this particular developmental phase in several mammal species, including humans, and to propose emerging hypotheses on the putative effect of estradiol signaling on the development and function of organs involved in female reproduction.

## 1. Introduction

Mini-puberty of infancy is an intriguing developmental period in mammals characterized by the postnatal activation of the hypothalamo-pituitary ovarian (HPO) axis (or gonadotrope axis) for a short duration. The term “mini-puberty” was first used in boys as the period of hormonal surge of gonadotropins and testosterone, which occurs in early infancy and contributes to the maturation of germ cells [1]. A wide array of studies has shown that both sexes display high levels of follicle-stimulating hormone (FSH) and luteinizing hormone (LH) and a gender-specific elevation of sex steroid hormones (testosterone in males and estradiol in females) postnatally. There is a sexual dimorphism in the surges of gonadotropins, with higher FSH levels in girls than in boys and higher LH levels in boys than in girls [2]. The postnatal elevation of testosterone levels in humans results in testicular descent, maturation of the testis, and penile growth [1,3]. It may also influence the masculinization of the brain, a process initiated during fetal life and mediated by testosterone produced by developing testes [3]. In male rodents, numerous studies, including the seminal work of Phoenix et al. on guinea pigs (1959) [4], showed that the perinatal period (end of gestation, first postnatal days) is critical for masculinization and defeminization of the nervous system. By contrast, mini-puberty has been far less studied in females, and its role remains elusive. Ovarian activity is particularly elevated after birth, with an intense synthesis and release of hormones, such as estrogens (estrone (E1) and estradiol (E2)). Their production depends upon the expression of steroidogenic enzymes, such as aromatase (CYP19A1), which ensures the conversion of androgens into estrogens. The balance between these biologically active metabolites, their precursors, and inactive sulfated sex steroids is controlled by the sulfotransferase (SULT)–sulfatase (STS) pathway (schematized in [5]). This notable ovarian activity occurs in time with the development of structures related to reproductive success. Estrogens may already act on several organs to influence their differentiation/maturation, particularly via the nuclear receptors ER (estrogen receptor), as discussed in this review. The conservation of this phase among female mammals suggests that this transient gonadotrope axis activation is important for reproductive function, but its role and regulation remain largely unexplored. In this review, we cover the literature on the activation of the HPO axis at mini-puberty, with a special focus on both the regulation of ovarian function and the potential role of this phase on reproductive success, mainly based on studies led in rodents. We will mainly discuss studies conducted in rodents since the literature on mini-puberty in humans has already been reviewed several times [6,7,8].

## 2. Defining HPO Axis Activation & Mini-Puberty Timing in Female Mammals

In female mammals, gonadotropins are already detected during fetal life. Their levels are schematized in mice at different periods of life (Figure 1; for humans [6,7]). This review will focus on their early regulation since the literature on this aspect of reproductive life, and reproductive senescence is abundant [9,10,11]. During fetal life, significant amounts of gonadotropins are detected (mouse: [12,13,14]; rabbit: [15]; human: [16,17]; rhesus monkey: [18]). LH and FSH start being detected at 16–17 days post-conception (dpc), i.e., 2–3 days before birth, in the female mouse [12,13] and at about 10–11 weeks of gestation in humans [17]. Their action on the fetal ovary may, however, greatly vary from one species to another. In humans and rhesus monkeys, gonadotropin signaling may become active by mid-term, as fetal ovaries are already a host of steroidogenesis with measurable androgens and estrogens. In addition, human fetal ovaries display growing follicles up to the antral stage at around five months of gestation [17,19,20,21]. The situation is different in rats and mice since, in these species, sex steroids become detectable after birth (around five days postnatal (dpn)) when gonadotropin receptivity and preantral follicles appear in the ovaries [22,23,24]. Gonadotropin levels then fall during perinatal life in humans [16,17,18,25,26], but not in mice [13].

The postnatal elevation of gonadotropin levels lasts for a few days in rodents and several months in larger mammals, including humans (Table 1). This is accompanied by a significant increase in ovarian activity, as shown by the elevation in estradiol levels. It is important to highlight that the duration of the mini-pubertal surges of gonadotropins and steroids may differ, with estradiol being much shorter (Table 1). The period of postnatal ovarian activity corresponds to mini-puberty [27,28,29,30,31]. Defining a precise time window of mini-puberty for a given species is difficult due to the considerable variability in gonadotropin/estradiol surges between individuals, the scarcity of longitudinal studies on the same subjects, and the poor reliability of commonly used hormone assays, in particular those for sex steroids. In girls, FSH levels increase at about one week of age and peak at 2–3 months, reaching values twice as high as those of women during the follicular or luteal phase [32]. They subsequently decline but remain higher than those of prepubertal girls until four years of age, remaining low up to puberty [33,34]. LH levels peak at around 1–3 months of age to decline after that, and unlike FSH, it does not reach levels above those of the menstrual cycle [32,33,35]. Estrone and estradiol are significantly produced by the human ovary during infancy [36] and can be detected in the serum [31,32]. Accurate determination of their levels by LC/MS-MS in children of different ages revealed that they are higher during the first three months of life (means: 35 pM and 10 pM, respectively) than in non-pubertal children (means: 10 pM and 3 pM, respectively), but they remain below those of the first days of the menstrual cycle (mean: 100 pM) [30].

In-depth analyses of the postnatal profiles of LH, FSH, and estradiol in rodents during the neonatal (0–7 dpn), infantile (8–17 dpn), and juvenile (17–30 dpn) periods (Figure 2A) and comparison with their profiles at puberty (occurring up to about 35 dpn in rats and 45 dpn in mice) and during reproductive life highlighted the tremendous activation of the gonadotrope axis during the infantile period in these species [27,29,30,37,38]. In rodents, the highest prepubertal FSH levels are reached at about 12–15 dpn, and we observed in mice that they are 3–5 times higher than at puberty or during reproductive life (Figure 2B) [27,29,30]. LH levels are elevated for about one week between 4 and 18 dpn, depending on rodent species and strains, and we found in mice that they reach values of the LH ovulatory surge at about 14 dpn (Figure 2B) [27,29,30,37,38]. Estradiol is detected in both the ovary and the serum as early as 5–7 dpn in mice and rats [29,30]. Circulating estradiol levels surge around 12–15 dpn, reaching preovulatory surge levels at 14 dpn in mice (Figure 2B) [29,30]. In contrast to the situation in humans where the “quiescence” of the HPO axis until the approach of puberty is relatively long (i.e., lasting for a minimum of 7 years), in rodents, there is a relatively short quiescence phase, lasting no more than 2–3 weeks.

## 3. Possible Mechanisms Underlying Increasing Gonadotropin Levels during Fetal Life and Mini-Puberty

### 3.1. Lack of Sensitivity to Estrogen Negative Feedback during Fetal Life

Gonadotropin-releasing hormone (GnRH) plays an important role in driving LH and FSH synthesis and release during the reproductive cycle. However, it may play an important role in regulating gonadotropins as soon as fetal life, as suggested by the onset of GnRH system activity (protein abundance, receptor binding, induction of gonadotropins) 2 to 3 days before birth in mice and at mid-gestation in humans, i.e., concomitantly with the observed increase in LH and FSH in the serum [12,42,43,44,45]. There is a marked sexual dimorphism in gonadotropin levels, which are generally much higher in females than in males (mouse: [12,13,14]; rabbit: [15]; human: [16,17]; rhesus monkey: [18]). In males, testes produce testosterone during fetal life, while in females, ovaries do not produce sex steroids (in rodents). This marked increase in gonadotropin levels in the female fetus may result from the lack of negative feedback from the ovary to the hypothalamus and pituitary. However, in rodents, as in other mammal species, the fetus is exposed to high levels of maternal steroids. In addition, in some species, including humans, sex steroids are produced by fetal ovaries [21]. Therefore, another explanation could be that the action of estrogens at the hypothalamic level in the fetus is prevented by estrogen-binding proteins, such as alpha-fetoprotein (AFP) in mice, precluding crossing of the brain blood barrier [46] (Figure 1). Noteworthy, AFP has a very low affinity for estrogens in humans, and thus the identification of binding protein(s) during fetal life remains to be investigated in this species.

What could explain the decrease in fetal gonadotropin levels observed around the term in humans? The extremely high concentrations of maternal estrogens may allow the generation of free circulating estrogens to cross the fetal brain-blood-barrier and target the hypothalamus and the pituitary. Observations suggest that pre-term human infants, not exposed to high maternal estrogens at the end of pregnancy display higher amplitude and duration of FSH surge after birth [47].

### 3.2. Limited Negative Feedbacks Exerted by Estrogens and Inhibins at Mini-Puberty

The progressive increase in circulating LH and FSH levels observed at mini-puberty may result from the progressive disappearance of maternal steroids from the circulation of the offspring. Why do FSH levels generally display a more dramatic elevation in amplitude and duration than LH levels in females? During reproductive life, it is assumed that high-frequency GnRH pulses promote LH release, while low-frequency GnRH secretion induces high levels of FSH secretion. Studies conducted on rat postnatal hypothalamic explants suggest that the frequency of GnRH pulses is low at mini-puberty and progressively increases during the prepubertal period [48]. Hence, one of the current hypotheses is that low-frequency GnRH pulses favor FSH secretion over that of LH at mini-puberty [49]. On the other hand, the picture may be far more complex. Indeed, even if LH levels remain relatively lower than those of FSH at mini-puberty, its levels are still high. Additional mechanisms regulate gonadotrope function, such as local factors and ovarian hormones. The possibility that this differential regulation of FSH and LH is driven by ovarian factors is supported by our work in mini-pubertal rats, showing that the anti-müllerian hormone (AMH) stimulates FSH secretion but not that of LH [50]. This marked elevation of FSH could also arise from the low mini-pubertal production of inhibin A and B, which would not be able to exert a negative action on FSH synthesis, as suggested by observations in the rat reporting a negative correlation between FSH levels and those of inhibins during the prepubertal period [30].

The elevation of gonadotropin levels could also result from the weakness of estradiol negative feedback due to high levels of AFP, which restrict the availability of free estrogens, at least in rodents. Since AFP levels decline markedly at mini-puberty to become low at around 20 dpn, when gonadotropin levels decrease significantly [51,52], the estradiol negative feedback system may become fully active [53,54]. However, there is some evidence that this feedback is functional despite high levels of AFP since ovariectomy in rats during mini-puberty further increases FSH levels, an effect that is prevented by estradiol supplementation [54]. Negative feedback may also exist in primates since bilateral gonadectomy in rhesus monkeys during infancy results in increased LH and FSH levels [55]. In addition, girls with Turner syndrome usually suffering primary ovarian insufficiency with low to undetectable levels of ovarian hormones, exhibit higher levels of FSH than healthy girls during mini-puberty [2]. Furthermore, studies conducted in human infants reported that higher gonadotropin levels at birth in pre-term girls than in full-term girls are associated with lower serum levels of estradiol and inhibin B [56]. Overall, these data indicate that regulating gonadotropins during infancy involves multiple factors acting at different levels of the HPO axis.

## 4. The Ovary: From Its Differentiation to Its Endocrine Activation at Mini-Puberty

The mini-pubertal ovary is endowed with many growing follicles in addition to containing primordial follicles. Unlike the adult ovary, follicular growth does not proceed beyond the antral stage (Figure 3). These early-growing follicles display very specific features, which are discussed in more detail below.

### 4.1. Follicle Formation

The formation of primordial follicles occurs after the fragmentation of ovigerous cords, which are structures containing oocytes arrested at the first prophase of meiosis and lined with pregranulosa cells. Depending on the species, this event occurs during fetal development (domestic mammals and humans) or around birth (rodents) [19,20,57,58]. However, in well-documented species such as rodents, fragmentation of ovigerous cords is reported to also give rise to primary-stage follicles, exhibiting enlarged oocyte size as well as cuboidal and proliferating granulosa cells [58]. Fragmentation of ovigerous cords starts from the center of the ovary at birth, yielding primary follicles, and progresses toward the periphery to give rise to primordial follicles around 4–5 dpn [59]. These primary follicles have never been in a dormant state, unlike primordial follicles. The emergence of these two populations of follicles may result from distinct waves of follicular cell differentiation, with the pregranulosa cells in primordial follicles migrating from the ovarian surface epithelium during perinatal life, whereas the granulosa cells in primary follicles would come from precursors from the coelomic epithelium of the bipotential gonad [60,61]. This concept of distinct waves of follicular cell recruitment to form two populations of follicles has also been proposed for sheep, in which a similar centrifugal pattern of follicular differentiation seems to occur [57]. It is unclear whether this also occurs in humans, where fragmentation of ovigerous cords spans from 19 to 35 weeks of gestation and primary follicles first appear around 21 weeks of gestation (for review [62]).

### 4.2. Roles and Dynamics of the First Follicular Waves

In rodents, primary follicles resulting from the fragmentation of ovigerous cords constitute the first follicular waves. They rapidly grow to the preantral and antral follicle stages at mini-puberty (Figure 3). The dogma that these first follicular waves are anovulatory and therefore degenerate through the process of atresia before puberty [58,63] has been challenged by two studies, including one from our group [64,65]. In a rat model only containing the first follicular waves following depletion of primordial follicles during the neonatal period (γ-irradiated rat model), it was shown that these growing follicles are progressively eliminated by follicular atresia but that a subset of them achieves their maturation to the ovulatory stage and contributes to the first sexual cycles of reproductive life (2 months of age) [64,66]. This follicular population entirely disappears by 4–6 months of age due to depletion by atresia and ovulation [64]. Similar findings were obtained in a more recent study conducted in a genetically modified mouse model, i.e., the *Forkhead box L2 (Foxl2)-CreER^T2^; mT/mG* mice, allowing fluorescent tracing of the first follicular waves and the primordial follicles over the prepubertal and adult periods [65]. This model also revealed that primordial follicles begin to be recruited to the growing follicle pool at approximately 13 dpn, i.e., at mini-puberty, thereby providing subsequent follicular waves contributing to ovulations during reproductive life from about three months of age. In humans, there is a long period between the appearance of the first growing follicles during fetal life and the onset of puberty (8–14 years), and it is, thus, unlikely that the growing follicles of the first follicular waves contribute to ovulation at puberty.

### 4.3. Mechanisms Controlling Intra-Ovarian Estradiol Production during Mini-Puberty

#### 4.3.1. Regulation by Gonadotropins

In female mice and rats, ovaries express FSH receptors (FSHR) in granulosa cells early as birth and LH receptors (LHR) in thecal cells from 5 dpn [24,67,68]. Similar to cyclic females, LH stimulates androgen synthesis by thecal cells, and FSH promotes the aromatase-mediated conversion of thecal-derived androgens into estrogens in granulosa cells [69]. FSH also regulates follicular growth during this period, as shown by the early arrest of folliculogenesis at the preantral/early antral stage in FSH receptor (*Fshr^−/−^)* and Fsh beta subunit (*Fshb^−/−^)* knock-out mice during the infantile period (Balla et al., 2003) [70]. Gonadotropins are essential for early follicular development, as shown by the phenotype of *hypogonadal* (*hpg*) mice, which display lower counts of preantral follicles than wild-type females and no antral follicles at 8 dpn [71]. From the γ-irradiated rat model, we specified that follicles from the first waves at the preantral/small antral stage express *Cyp19a1* aromatase (the enzyme responsible for the conversion of androgens produced by thecal cells into estrogens by granulosa cells), and are responsible for the synthesis of estradiol throughout the prepubertal period [64]. Their granulosa cells also express *Lhcgr* (LHR) [29,64] (Figure 4). These findings indicate that these small follicles display premature endocrine maturation since *Cyp19a1* and *Lhcgr* are detected in granulosa cells at the large antral/preovulatory follicles in the adult ovary [72]. While the functionality of LHR signaling in granulosa cells remains to be demonstrated in mini-pubertal ovaries, it is likely that FSHR signaling stimulates estradiol production by inducing the expression of *Cyp19a1* [29,64,70]. In vitro analyses on isolated preantral follicles from 14 to 17 dpn (mini-pubertal) mouse ovaries confirmed the presence of *Fshr* and *Cyp19a1* transcripts [73]. Paradoxically, despite the mini-pubertal elevation of both FSH and LH levels at mini-puberty and the notable production of estradiol, follicles do not grow beyond the antral stage [29]. By manipulating gonadotropin levels in vivo in the mouse with a GnRH-R antagonist, we could demonstrate that high FSH levels up-regulate the expression of both *Lhcgr* and *Cyp19a1* and estradiol production from the first follicular waves. However, they are inefficient in inducing the expression of the cell cycle promoter Cyclin D2 (*Ccnd2*) and stimulating follicular growth [29]. Contrary to these findings, in vitro FSH treatment of preantral follicles from mini-pubertal mice induces both *Cyp19a1* and *Ccnd2* expression, and it has a follicular growth-promoting action [73]. We hypothesize that these contradictory data may result from differences in experimental settings (in vitro versus in vivo), on the type of used FSH (recombinant human FSH in Hardy et al., 2017 [73] versus pituitary-purified ovine FSH in François et al., 2017 [29]) and on FSH concentrations (10–100 ng/mL in vitro in Hardy et al., 2017 [73] to 150 ng/mL in vivo and 500 ng/mL ex vivo in François et al., 2017 [29]). The preferential action of elevated FSH levels on the steroidogenic pathway observed in vivo is further illustrated by the fact that treatment of mini-pubertal mice with exogenous gonadotropins following a superovulation procedure further increases estradiol levels without promoting preantral/early antral follicle growth [29]. Taken together, we hypothesize that gonadotropins stimulate estradiol synthesis in the first follicular waves only when they reach high concentrations, which are inefficient for follicular growth [29] (Figure 5). This mechanism may prevent premature ovulation.

#### 4.3.2. Contribution of AMH Signaling

AMH is expressed by granulosa cells as soon as the primary follicle stage, and it disappears in antral follicles (except in cumulus cells) and atretic follicles [74]. AMH may inhibit the recruitment of primordial follicles into the growing pool and decrease the acquisition of FSH receptivity by antral follicles [75]. In mouse and rat ovaries, *Amh* expression in the first follicular waves progressively declines during mini-puberty, and its expression is essentially ensured by subsequent follicular waves located in the periphery [66,76]. We found that *Amh* down-regulation resulted from the mini-pubertal surge in FSH levels [76]. Similar to the adult ovary, AMH down-regulates *Cyp19a1* expression during mini-puberty [76]. Therefore, we hypothesize that FSH-induced down-regulation of AMH facilitates FSH-promoting action on estradiol biosynthesis by the first follicular waves [76], as described in the adult ovary [75,77,78]. Indeed, FSH-induced repression of *Amh* expression was observed in preantral follicles collected from mouse mini-pubertal ovaries and cultured in vitro [73]. The same may be true in mini-pubertal ewe lambs, where females with high FSH levels show low circulating AMH levels, while those with low FSH levels exhibit high AMH levels [39]. As highlighted above, AMH could also exert extra-gonadal actions, at least in rodents. Indeed, it could enhance GnRH neuron activity in the hypothalamus and LH secretion from the pituitary in adult females [79]. It could also directly act on the pituitary to increase FSH production and secretion in mini-pubertal females [50]. Although the origin of AMH acting on the hypothalamus and the pituitary remains elusive, it is tempting to suppose that ovarian AMH contributes to the activity of the gonadotrope axis during mini-puberty (Figure 5).

#### 4.3.3. Effects of Exposure to Endocrine Disruptor Chemicals (EDCs) on Mini-Pubertal Estradiol Production

The potential impact of exposure to EDCs, specifically at the time of mini-puberty, remains largely unexplored. However, several studies have assessed developmental periods of exposure ranging from fetal to postnatal periods, including mini-puberty. The presence of metabolites of EDCs such as phthalates in maternal urine or cord blood, possibly lowering testosterone production, is associated with shorter anogenital distance and cryptorchidism in boys (for review, [80]). In girls, the relatively subtle changes in physical parameters during mini-puberty, and the need for more invasive investigations, may complicate studying EDC effects. However, assessment of pubertal markers in girls showed associations between phthalate or bisphenol A (BPA) metabolites and altered age of pubertal onset [81,82].

In recent studies carried out in rodents, we compared the impact of gestational/lactational versus mini-pubertal exposure to EDCs such as 2,3,7,8-tetrachlorodibenzo-*p*-dioxin (TCDD, the “Seveso dioxin”) reported to interfere with ER signaling or steroid synthesis upon binding to the aryl hydrocarbon receptor (AHR). TCDD affects the ovarian expression of *Cyp19a1* and estradiol production in juvenile and adult ovaries [83,84,85]. It induces the expression of detoxifying enzymes, for instance, *Cyp1a1,* to metabolize xenobiotic substances [86]. Studies in rodents exposed to TCDD in utero and during lactation up to weaning at around 20 dpn show that this compound causes delayed puberty, early alteration of estrous cycles, and impaired fertility [87]. Interestingly, in utero and lactational exposure to TCDD of female rats (gavage with 200 ng kg^−1^ BW of the mother at 15.5 dpc) has no effect on the expression of steroidogenesis-related factors in the ovary during mini-puberty, although it induces AHR detoxifying machinery [88]. Similarly, intra-peritoneal TCDD injection of mini-pubertal mice at 13 dpn (5 mg kg^−1^ BW) has no effect on steroidogenesis, but it activates the AHR detoxifying pathway in the mini-pubertal ovary [89]. These observations contrast with the fact that administration of the TCDD in mice at the approach of puberty stimulates estradiol synthesis in addition to activating the detoxifying pathway [89].

These findings suggest that although TCDD may activate AHR signaling during mini-puberty, it does not affect ovarian endocrine activity at this stage. Interestingly, even the endogenous AHR pathway is unable to regulate estradiol synthesis during mini-puberty, unlike peripubertal mice, as suggested by studies on *Ahr* knock-out (*Ahr^−/−^*) mice [89]. These studies suggest that AHR-mediated EDC action depends on the stage of sexual maturity, as proposed [85]. However, this does not rule out that EDCs to which infants are exposed during mini-puberty, acting via AHR-independent mechanisms such as phthalates and BPA, may impact ovarian activity.

## 5. Putative Physiological Roles of Estradiol during Mini-Puberty

### 5.1. Developmental Organization of the HPO Axis and Female Behavior

During most of the cycle, low levels of estradiol suppress gonadotropin secretion through negative feedback exerted on the hypothalamus. In contrast, at the end of the follicular phase (proestrus in rodents), high estradiol levels trigger positive feedback that induces a large and continuous increase in GnRH release. This GnRH “surge,” with an increase in gonadotrope cell responsiveness to GnRH, causes a surge in LH release from the pituitary. This ovulatory surge of LH occurs in all female mammals and is necessary for ovulation and sexual receptivity in rodents. The effects of estradiol on the regulation of GnRH secretion, either negative or positive, are integrated into the hypothalamus at the level of kisspeptin neurons, which express sex steroid receptors. These neurons represent major gatekeepers of pubertal onset and female reproduction (for review [90]. They send projections to cell bodies and terminal nerves of GnRH neurons, thereby activating both GnRH synthesis and liberation [91,92]. There are two kisspeptin neuronal populations in the murine hypothalamus, located in the rostral periventricular area of the third ventricle (RP3V) of the preoptic area (POA) and the arcuate nucleus (ARC). The RP3V population, which integrates the positive estradiol feedback to induce ovulation, is sexually dimorphic, with more kisspeptin neurons in females than in males [92].

The high levels of progesterone-induced by the preovulatory estradiol surge followed by the ovulatory LH surge activate the neural circuitry underlying female sexual behavior, including the receptive posture, called lordosis [93]. This circuitry includes the olfactory system and chemosensory areas, including the medial amygdala, the bed nucleus of stria terminalis, and the ventromedial hypothalamus, which processes the chemosignals into behavioral responses. All of these structures express high levels of ERα, with particularly high progesterone receptor expression in the ventromedial hypothalamus.

As reported above, the negative feedback system from the ovary to the hypothalamus may already be functional at birth. In contrast, the positive feedback loop may not be functional before puberty, possibly because POA kisspeptin neurons are not fully developed or not yet included in a functional network with GnRH neurons. Nevertheless, prepubertal ovarian estrogens exert a developmental control of neural structures involved in HPO axis regulation and female behaviors. Indeed, female mice with deletions in the *Cyp19a1* gene exhibit fewer hypothalamic kisspeptin neurons in the POA at 20 dpn and reduced lordosis behavior in adulthood [94,95,96]. Administration of estradiol to these knockout females during the infantile/juvenile period (15–25 dpn) partially restores these effects [96,97]. In addition, postnatal OVX at 15 dpn reduces the number of kisspeptin neurons in the POA, and adult estradiol supplementation restores it [98]. Studies of these and other experimental models have led to the idea that the period around 15 dpn in mice may be a critical window for exposure to estradiol on kisspeptin neurons, the age at puberty, and sexual behavior [98] (Figure 6). The possibility of an early estradiol action in females is further supported by the observation that ERα is expressed from birth (mRNAs and protein levels) in brain areas that will regulate the HPO axis and behaviors in adults [99,100,101]. In addition, genetic studies have shown that estrogens produced before puberty act through neural ERα and ERβ signaling pathways. Early embryonic *Esr2* deletion in the nervous system delays kisspeptin expression in prepubertal females and the onset of puberty [102]. Furthermore, early *Esr1* deletion in kisspeptin neurons or the nervous system accelerates pubertal onset, suppresses cyclicity and sexual behavior, and leads to an infertile phenotype [103,104].

### 5.2. Mammary Gland Development

The development of the mammary gland, a process referred to as mammogenesis, occurs during embryogenesis by the invagination of ectodermal structures to form epithelial buds surrounded by multi-layered mesenchymal cells. Epithelial cell proliferation and branching lead to the formation of a small ductal tree at birth. In mice, very little growth occurs between 11.5 and 16.5 dpc, but during the last few days of gestation, there is rapid proliferation of mammary epithelial cells, accompanied by penetration into the mammary fat pad precursor tissue. The involvement of ovarian hormones in mammary development was proposed from the observed sex dimorphism between males and females. While early mammogenesis occurs in the absence of gonadal sex steroids in the female mouse [105], it would be stimulated by high levels of maternal steroids in late gestation when there is rapid mammary gland growth. In males, the production of testosterone by the fetal testis leads to the apoptosis of the underlying mesenchymal cells that express the androgen receptor, and mammary glands remain rudimentary throughout life. In female mice and rats, extensive proliferation and branching of the mammary duct system occur between birth and puberty. Mammary gland growth is isometric (meaning that it grows at the same rate as the body) for 2 to 3 weeks, beginning around 7 dpn, and it becomes allometric (meaning that it grows faster than the other parts of the body) at puberty. This postnatal growth results in the development of a large number of end buds; in mice but not in rats, it could be estradiol-dependent, especially at puberty [106]. The binding of estradiol and expression of ER is observed in mammary epithelial cells during mini-puberty in the mouse [107,108]. However, mammary gland development proceeds to some extent in the absence of estradiol since the conditional loss of ERα in mammary epithelial cells arrests mammary gland development at the prepubertal stage [109]. However, in the mouse, most investigations have been carried out during the juvenile and pubertal periods, partly due to the prevailing view that estradiol production by the ovaries is negligible prior to puberty and that high levels of AFP prevent estradiol action [107]. Unlike mice, the role of postnatal estradiol in mammogenesis seems to be confirmed in cattle and goats, which show positive allometric mammary gland growth at the time of mini-puberty [110,111,112,113] (Figure 6). ERα expression is already detected in mammary epithelial cells at three months in calves [114]. Ovariectomy of mini-pubertal heifers and goats severely impairs the subsequent development of mammary glands, showing poorly organized epithelial structures, low cell proliferation and tissue remodeling [111,113]. In humans, ERα expression is observed in epithelial cell nuclei from the 30th week of gestation onwards, and it is markedly up-regulated shortly after birth [115]. The observation supports the postnatal involvement of ovarian hormones in mammary gland development that mammary gland diameter decreases two months after birth in boys but not in girls and the fact that pre-term girls, who have higher levels of estradiol than full-term girls, one month after birth, transiently display enlarged mammary glands during mini-puberty [116]. In addition, a 6-week supplementation starting at the birth of extremely premature infants (mean delivery age: 26 weeks) with estradiol and progesterone leads to a dramatic increase in mammary gland diameter 4 to 5 weeks after birth [117].

### 5.3. Uterus Development

The uterus consists of two compartments, the myometrium and the endometrium, the inner mucosal lining of the uterus. The endometrium contains different cells, i.e., epithelial, stromal, immune, and endothelial cells. The epithelium is divided into two types: the luminal epithelium and the glandular epithelium forming the uterus glands and displaying an important role in supporting pregnancy by producing substances essential for the development and survival of the conceptus. The primitive uterus, also called Müllerian ducts, is not fully developed at birth in mammals, including cattle, humans, and rodents. Additional steps, such as differentiation and growth of the myometrium and epithelial glands, take place postnatally, including during mini-puberty.

The role of ovarian hormones and ER signaling on uterus development has been widely studied and may be species-dependent. In rodents the uterus grows and develops normally in the absence of ovaries following neonatal ovariectomy, but only for the first three weeks after birth. The role of ovarian factors in subsequent uterus development is shown by observing impaired uterine growth and maintenance of uterine glands in neonatally ovariectomized females [118,119,120]. On the other hand, when these females are supplemented with estradiol or progesterone during the neonatal period, their uterus develops normally past three weeks of age, implying that early exposure to sex steroids may have a long-term effect on this organ. ERα is expressed as early as 5 dpn in epithelial and stromal cells in mice, but it shows a stronger expression in stromal cells [121]. Tissue recombination studies indicate that epithelial ERα is neither necessary nor sufficient to mediate the mitogenic action of estrogens on the epithelium. This would be primarily mediated by stromal ERα via the production of stromal-derived growth factors such as epidermal growth factor (EGF) and insulin growth factor (IGF) 1 and 2 [122,123]. Adult mice lacking ERα (*Esr1*-null) have hypoplastic uteri containing all cell types in reduced proportions, but fetal uterine organogenesis is normal [124,125]. In these mice, the initial stages of uterine development and gland formation are normal from birth to 22 dpn, and these processes are subsequently affected [120,126]. Importantly, when neonatal uteri of *Esr1*-null mice are grafted into adult ovariectomized mice, estradiol stimulates graft growth only in wild-type hosts, suggesting that this hormone stimulates the production of a systemic growth factor of as yet unknown origin that would rely on extra-uterine ERα-dependent mechanisms to promote uterine growth and gland formation [122]. Taken together, these data indicate that estradiol would regulate uterine growth and development by intra- and extra-uterine ERα-dependent signaling primarily from the juvenile period in mice.

In sheep, ERα is already expressed during mini-puberty in several cell types of the uterus, namely glandular epithelium, stromal cells, and myometrial cells [127]. Neonatal ovariectomy reduces uterus growth and the number of glands observed at 56 dpn (mini-puberty) [128]. In addition, in *Inverdale* ewes carrying a natural homozygous mutation in the bone morphogenetic protein 15 (*BMP15*) gene, impaired ovarian function is associated with a significant reduction in uterine growth and fewer uterine glands [129]. These data imply that ovarian hormones play a role in this process. However, estradiol is unlikely to contribute to all the developmental and differentiation processes since the administration of an anti-estrogen (ERα antagonist) from birth to 55 dpn alters endometrial glands and ductal gland invaginations, but it does not affect uterus growth [127]. The deleterious effect of the exposure to supra-physiological doses of estradiol is supported by the observations that repeated injections of large amounts of estradiol-valerate from birth to 55 dpn inhibit uterine growth and endometrial gland differentiation, possibly by repressing ERα expression [127]. In humans, ERα mRNAs are detected in the uterus at 19 weeks of gestation, but the cell type expressing this receptor has not been reported [130]. Uterine length increases in pre-term girls compared with full-term girls at one month of age, positively correlated with urinary estradiol levels [116]. Additional evidence that the uterus is already the target of sex steroids after birth is provided by the observation that a 6-week replacement of estradiol and progesterone in pre-term infants dramatically enlarges the size of the uterus related to the control group (not replaced) group [117]. Taken together, these studies suggest that mini-pubertal ovarian activity, possibly through estradiol action, may regulate uterus development in some species during mini-puberty (Figure 6).

## 6. Concluding Remarks, Perspectives and Future Directions

Although recent investigations on mini-pubertal girls support the idea that this developmental phase has an important role in programming fertility by acting on different levels of the reproductive system, some key experimental clues have been provided by animal models, which will probably provide additional evidence in the future. As reported in this review, estrogens appear necessary for the maturation of the Kiss-GnRH system in the hypothalamus to modulate, *in fine*, the cyclic regulation of LH as well as sexual and maternal behaviors. They could also influence the differentiation of both the mammary gland and the uterus, probably in a species-dependent manner. Therefore, the postnatal endocrine activity of the ovary could be of major importance for subsequent fertility. A better understanding of the physiological role of this period and the regulation of the HPO axis is essential to determine whether this could be a potential time window of vulnerability, as suspected. Indeed, a visible advance in the age of puberty has been observed in recent years in girls, and it is estimated that 6–20% of reproductive-aged women have fertility defects due to polycystic ovarian syndrome (PCOS). Although the etiology of these defects remains uncertain, it has been suggested that early disturbances of the sex steroid system during fetal and postnatal life by daily exposure to EDCs with estrogenic, anti-estrogenic, or anti-androgenic activities could be involved [131,132]. The possible deleterious impact of EDCs on HPO activity and the differentiation of estradiol target organs during mini-puberty has yet to be assessed to establish their long-term impacts on reproductive health.

## Figures and Tables

**Figure 1 ijms-23-13695-f001:**
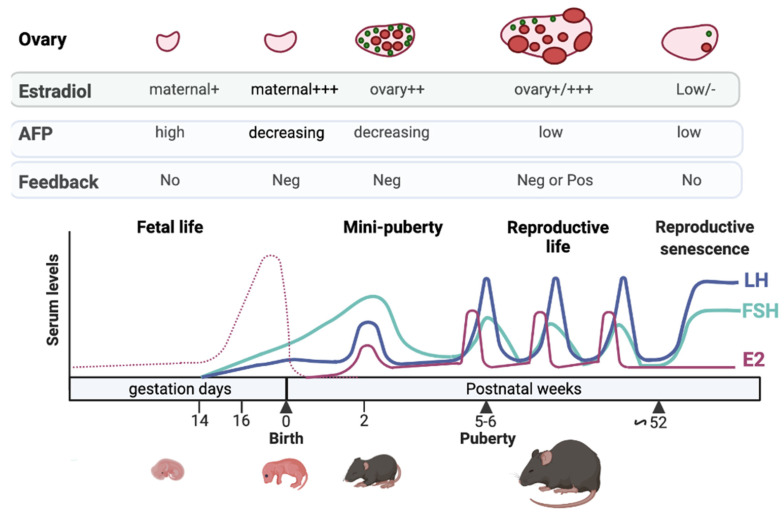
Regulation of the gonadotrope axis at different periods of life in the mouse. FSH (green line) and LH (blue line) are detected in the last part of gestation, induced by GnRH stimulation. The high levels of maternal estrogens (dotted pink line) may not exert negative feedback at this time on FSH and LH synthesis and secretion due to high levels of α-fetoprotein (AFP) produced by the fetal liver. After birth, maternal estradiol wanes from pup serum, and this may lead to the loss of estradiol negative feedback and to the dramatic rise in gonadotropin levels observed at the time of mini-puberty. Additional ovarian factors may contribute to this rise (see the text). The gonadotropin surge contributes to increased estradiol production by the ovary at mini-puberty (continuous pink line). After that, the decrease in AFP after birth, and the maturation of inhibin negative feedback, may contribute to the observed fall in gonadotropin levels, remaining low up to puberty. During reproductive life, gonadotropin levels are regulated by both estradiol negative and positive feedback. Estradiol negative feedback in aged mice is attenuated, thereby increasing gonadotropin levels. Note that the levels of hormones shown during the different periods are given on an indicative basis since no studies encompass all these periods. See references in the text. The figure was prepared with the help of Mr. Le Ciclé, using BioRender, under Academic License terms.

**Figure 2 ijms-23-13695-f002:**
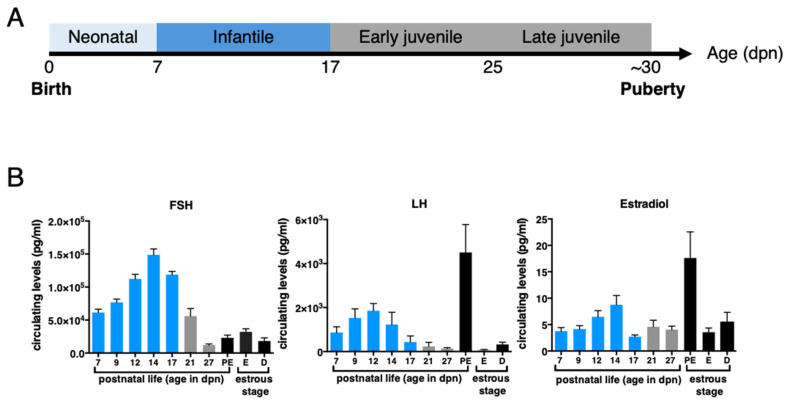
Significant changes in reproductive hormone levels during prepubertal life in the mouse. (**A**) The different stages of the prepubertal period in the mouse. The chronological axis displays the neonatal, infantile, and juvenile periods in this species. Mini-puberty takes place within the infantile period. (**B**) Gonadotropin and estradiol levels in prepubertal and adult female mice at different stages of the estrous cycle were determined by Luminex assay and GC/MS-MS, respectively, as published by our group in [29], including the additional age of 27 dpn analyzed at the same time as the other ages using the same methods. Hormonal levels of the infantile period are shown as blue bars.

**Figure 3 ijms-23-13695-f003:**
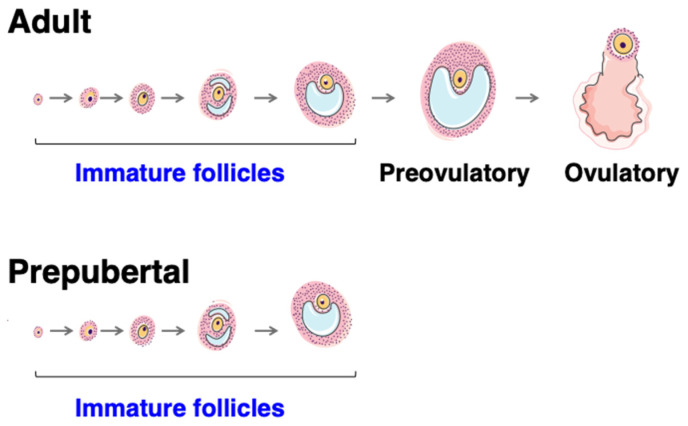
Different categories of follicles are present in the prepubertal ovary in mammals. The prepubertal ovary is endowed with growing follicles up to the antral stage, and unlike the adult ovary during reproductive life, it does not display large healthy follicles of the size of preovulatory follicles despite high gonadotropin levels during mini-puberty.

**Figure 4 ijms-23-13695-f004:**
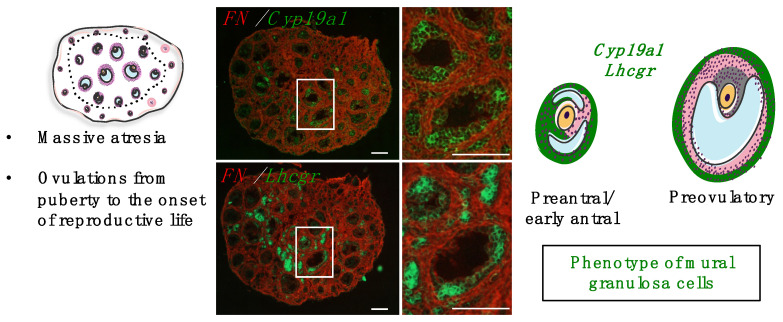
Particular characteristics of the first follicular waves in rodents. The first growing follicles are located in the central region of the ovary, delimited by dotted grey lines. Although most are destined to follicular atresia before puberty, a subset is ovulated at puberty and the very beginning of reproductive life [64,65]. The first follicular waves would be responsible for ovarian endocrine activity throughout prepubertal life [64]. During mini-puberty, immature follicles at the preantral/early antral stage show some functional characteristics of preovulatory follicles, such as *Cyp19a1* and *Lhcgr* expression in granulosa cells [29]. Scale bars: 100 μm.

**Figure 5 ijms-23-13695-f005:**
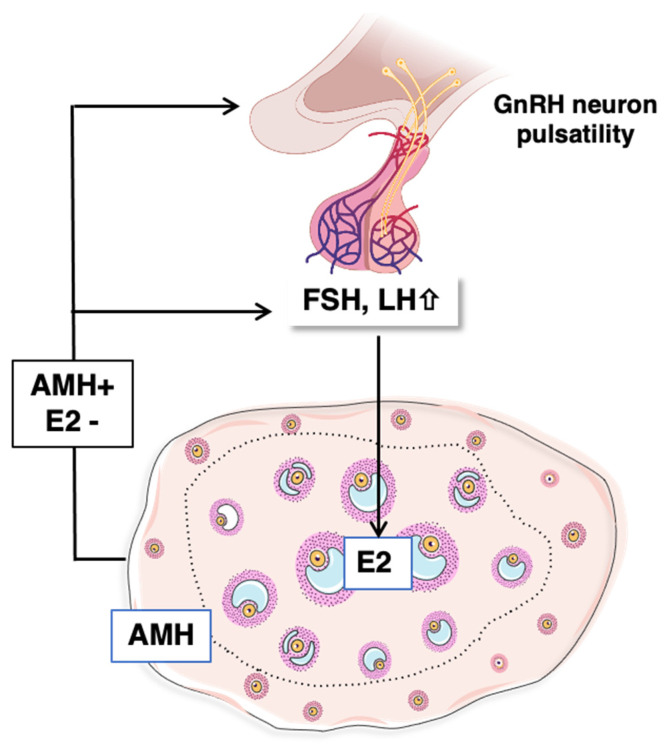
Proposed hypothesis on the dialogue between the ovary and the hypothalamo-pituitary system at mini-puberty. Estradiol produced by the first follicular waves may exert a negative feedback loop on the hypothalamus to restrain gonadotropin synthesis and secretion; however, the high levels of AFP may still limit its action. In contrast, we hypothesize that AMH produced by the subsequent follicular waves may up-regulate FSH synthesis by the pituitary and possibly LH and FSH levels by acting on GnRH neurons in the hypothalamus. See references in the text.

**Figure 6 ijms-23-13695-f006:**
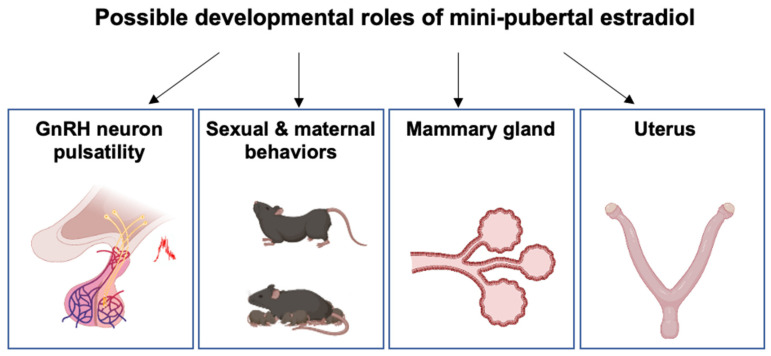
Possible roles of mini-pubertal estradiol on different organs related to reproduction. This hormone could play a pleiotropic role in the body by regulating the development and differentiation of several target tissues, including the hypothalamic system driving GnRH pulsatility, several brain areas contributing to sexual and maternal behaviors (amygdala, olfactory system, hypothalamus), the mammary gland and the uterus. See references in the text.

**Table 1 ijms-23-13695-t001:** Approximate ages of HPO activation and puberty in humans and different mammal species. ND: not determined.

Species	Ages at Gonadotropin Surge	Ages at Estradiol Surge	Ages at Puberty
**Mouse**	FSH: 9–17 dpn; LH: 9–14 dpn [29] FSH, LH: 4–18 dpn [37] FSH: 11–16 dpn; LH: 11–14 dpn [38]	12–14 dpn [29]	28–45 dpn
**Rat**	FSH: 1–17 dpn; LH: 11–21 dpn [27] FSH: 10–20 dpn; LH: 10–25 dpn [30]	9–19 dpn [27] 10–20 dpn [30]	28–35 dpn
**Sheep**	FSH: 2–7 weeks [39] LH: ND	ND	25–35 weeks
**Cow**	FSH, LH: 2–14 weeks [40]	ND	6–20 months
**Chimpanzee**	FSH, LH: 1–4 months [33] FSH, LH: 0.1–5 months [41]	ND	8–12 years
**Rhesus Monkey**	FSH: 0.5–4 months LH: ND	ND	2.5–3 years
**Human**	FSH: 1–4 years, LH: 1–3 years [33] FSH, LH: 1 year [32] FSH, LH: 0.5–4 years [34]	6 months [32] 2–3 months [28,31]	8–14 years

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
