# Peer review of "Deciphering the Roles & Regulation of Estradiol Signaling during Female Mini-Puberty: Insights from Mouse Models"

_ijms, 2022, doi:10.3390/ijms232213695_

Round 1
Reviewer 1 Report (Previous Reviewer 1)
Dear the Editor
This revised manuscript appears to be fine. In particular, Figures are improved and look attractive for the audience of this journal. In section 4, the signaling mechanism of FSH-mediated development was described in detail. The mechanism would help understanding the follicular development in adult and prepubertal state, even if the model animals were studied.
Minor concerns:
"Deciphering the roles & regulation of estradiol signaling during femalemini-puberty: insights from animal models" would be better to be:
"Deciphering the roles and regulation of estradiol signaling during femalemini-puberty: insights from animal models" .
Author Response
We thank the reviewer for his/her appreciation of our review and for the fact that he/she acknowledges the interests of this review to the audience of IJMS. Below are our point-by-point answers to his/her comments :
The title of the previously submitted manuscript was: “Roles & regulation of estradiol signaling during female mini-puberty: insights from animal models”. As requested by reviewer 1, and considering the remarks of reviewer 2 that the reviewed litterature is mainly on the mouse, we have modified the title to: “Deciphering the roles and regulation of estradiol signaling during female mini-puberty: insights from mouse models”.
Reviewer 2 Report (New Reviewer)
The manuscript authored by Marie M Devillers et al. describes Roles & regulation of estradiol signaling during female mini-puberty: insights from animal models.
Comments:
A thorough revision of the language should be carried out. Also, the editing of the document should be reviewed since there are different fonts in the text.
Part of the title refers to "animal models"; however, the animal on which most of the text is based is the mouse. In fact, this animal is mentioned approximately 28 times in the text and the other species (cow, sheep, etc.) are rarely (some of them only once) mentioned. It is important to point out that there are articles published in the literature since the 1980s (swine, cattle, etc.) up to the present day where each of the sections included in this work is conveniently dealt with. Therefore, if there is not a profound modification of the text, a change in the title should be made where "mouse model" is expressed instead of "animal models".
On the other hand, in Table 1, in the column referring to “ages at puberty”, in bovines, chimpanzees, and rhesus monkeys, only one value is referred to. However, as in the other species included, a time interval (minimum and maximum values) should be indicated. Taking into account that the age of puberty is a variable parameter in all animal species, the information on the above-mentioned species should be as indicated.
In conclusion, the submitted paper should be thoroughly reviewed, edited, and modified before being accepted for publication.
Author Response
We thank the reviewer for critical reading of our review and for his/her suggestions. Below are our point-by-point answers to his/her comments:
- As requested, we have extensively edited all parts of the manuscript. Changes made in the text are marked in yellow. The fonts were also reviewed. We believe that in its present form, the review is easier to understand.
- Following the reviewer's point that most of the reviewed literature is based on the mouse, we made the requested modification in the title by changing “animal models to “mouse models”. Please note that we also included the change requested by the other reviewer, so that the title of the manuscript is now: “Deciphering the roles and regulation of estradiol signaling during female mini-puberty: insights from mouse models”.
- We agree with the reviewer that time interval at puberty should be given in all the species shown in Table 1. Chimpanzee puberty occurs at approximately 8 to 12 years (Am J Primatol. 2020 Nov;82(11):e23064. doi: 10.1002/ajp.23064. by Kris H Sabbi et al), and that of macaques at 2.5-3 years (https://primate.wisc.edu/primate-info-net/pin-factsheets/pin-factsheet-rhesus-macaque/). Regarding the cow, we found that puberty could extremely vary depending on cattle breed, from 8 months to 20 months (for more information, please see https://www.sites.ext.vt.edu/newsletter-archive/livestock/aps-04_03/aps-315.html). We have added the informations for these different species in Table 1.
Round 2
Reviewer 2 Report (New Reviewer)
Dear Editor, please accept the manuscript in the present form.
This manuscript is a resubmission of an earlier submission. The following is a list of the peer review reports and author responses from that submission.
Round 1
Reviewer 1 Report
Dear the Editor
Devillers MM et al nicely summarized the role of estradiol on female development at younger period. Overall, this manuscript seemed well organized and covered most area of this field.
A major concern:
Estradiol and its related compounds are regulated genetically as described in this manuscript. In addition, sulfatase and sulfotransferase are other relevant factors for this metabolism. A short note for this may help readers consult further references. A concise figure for this may also be helpful.
Author Response
We thank the reviewer for his/her enthusiastic appreciation of our review. Below is our response:
As requested, we have included the suggestion about sulfatase and sulfotransferase (lines 38-43), and we have added the reference of a review recently published in IJMS on this subject (Tofovic & Jackson Int. J. Mol. Sci. 2020, 21(1), 116; https://doi.org/10.3390/ijms21010116).
Reviewer 2 Report
In this review, Devillers and colleagues summarized the mini-puberty in female mammals. However, quality of the manuscript is entirely very low. The authors must totally re-write the manuscript with attention to the following points.
1. There are so many improper words and grammatical errors. English editing by native speakers is necessary.
2. In HPO axis, H means hypothalamic. Such fundamental errors are big problems.
3. Line52-What is a gonadotropin levels? This sentence is difficult to understand for general readers.
4. Line89- It is strange that mice puberty is later than rat one.
5. Line153~156- Such invalid hypothesis is improper in review paper.
6. 3.2(Line155~197)-Again, this is an invalid hypothesis.
7. Section 4 and 5 are disorganized and boring. Almost parts are general summary for organ development, not related to mini-puberty.
8. Almost parts of concluding remarks did not reflect the previous parts. This is not a conclusion.
Author Response
Below are our comments on the different issues raised by the reviewer, in a point-by-point response:
- There are so many improper words and grammatical errors. English editing by native speakers is necessary
The two senior authors routinely use English language, notably through the preparation and review of scientific articles. The manuscript has nevertheless been re-edited and word/grammatical corrections are marked in yellow.
- In HPO axis, H means hypothalamic. Such fundamental errors are big problems
The criticism of the reviewer is not clear. Does he/she mean that we should not refer to the hypothalamus in the axis ? Use of the term “hypothalamo-pituitary-ovarian” axis is very casual in our field. Activity of the hypothalamus is key to the function of the pituitary, and not to be demonstrated. Besides, our review includes information on the hypothalamus.
- Line52-What is a gonadotropin levels? This sentence is difficult to understand for general readers
Indeed, this sentence was not clear and it has been edited (lines 54-56): “In female mammals, gonadotropins are not only present during reproductive cycle and senescence (corresponding to the cessation of cyclicity), they are detected as soon as the fetal period.”
- Line89- It is strange that mice puberty is later than rat one.
This difference has been largely published (see for instance the publication from Tena-Sempere’s lab: Gaytan et al, Sci Rep. 2017 Apr 12;7:46381. doi: 10.1038/srep46381).
- Line153~156- Such invalid hypothesis is improper in review paper.
Our hypothesis cannot be “invalid “as stated by the reviewer. Indeed, the definition of a hypothesis is the following: “a supposition or proposed explanation made on the basis of limited evidence as a starting point for further investigation, without any assumption of its truth”. The currently proposed hypothesis for the fetal decrease in gonadotropin levels at term in humans is formulated with caution (“we speculate “) and a reference (Kuiri-Hanninen et al, 2011). The reviewer does not give any cue to comment his/her point.
- 3.2(Line155~197)-Again, this is an invalid hypothesis.
Again, the reviewer does not give any reference to comment his/her point, and as commented in bullet 5, a hypothesis cannot be invalid without investigations by experimental approaches. The formulated hypothesis is based on works of renowned experts in the field and is shared by several investigators in the field.
- Section 4 and 5 are disorganized and boring. Almost parts are general summary for organ development, not related to mini-puberty.
We do not share the reviewer’s view. These two sections give a clear state-of the art view of the subject and provide the tools required to understand the physiology of the HPO axis at mini-puberty and its possible roles. This is to our knowledge, the first time that such a review, gathering information related to early HPO development and its possible consequence in physiology, is written.
- Almost parts of concluding remarks did not reflect the previous parts. This is not a conclusion.
The present conclusion opens new perspectives on researches related to mini-puberty by developing pathophysiological aspects. However, we have extended the conclusion by summarizing the main sections of the review (lines 494-498).
Round 2
Reviewer 2 Report
It is compelled to feel that revised version is not acceptable. There are many strange points. I wonder the senior authors carefully checked the manuscript. For example,
Line56-'please see'; Such words are hardly seen in the papers.
Line 45, 496; What is 'differenciation'?
One sentence is too long in total part.
Conclusion part includes many irrelevant sentence. There are various descriptions that appeared at first time in this section. These are not conclusion.
Author Response
Below is our point-by-point response to the reviewer. We have modified the minor spelling errors raised by the reviewer.
It is compelled to feel that revised version is not acceptable. There are many strange points.
I wonder the senior authors carefully checked the manuscript.
Line56-'please see'; Such words are hardly seen in the papers.
We have removed “please see” (line 56).
Line 45, 496; What is 'differenciation'?
We have corrected the typo found in “differenciation” and written “differentiation” instead (lines 45, 509).
One sentence is too long in total part.
What is the sentence in question?
We have also modified a few words or sentences, marked in yellow in the text.
Conclusion part includes many irrelevant sentence. There are various descriptions that appeared at first time in this section. These are not conclusion.
We followed the reviewer’s suggestion by remodeling the conclusion to reflect more closely the content of the review (lines 513-520). We have renamed this part as “Concluding remarks, perspectives and future directions » since in addition to providing an overall view of the topics, it proposes that EDC exposure during mini-puberty may result in reproductive health diseases.